# High-resolution genetic mapping reveals *cis*-regulatory and copy number variation in loci associated with cytochrome P450-mediated detoxification in a generalist arthropod pest

Seyedeh Masoumeh Fotoukkiaii[1], Nicky Wybouw[2,3], Andre H. Kurlovs[2,4], Dimitra Tsakireli[5,6], Spiros A. Pergantis[7], Richard M. Clark[4,8], John Vontas[5,6], Thomas Van Leeuwen[2]*

1 Department of Evolutionary and Population Biology, Institute for Biodiversity and Ecosystem Dynamics, University of Amsterdam, Amsterdam, the Netherlands, 2 Laboratory of Agrozoology, Department of Plants and Crops, Faculty of Bioscience Engineering, Ghent University, Ghent, Belgium, 3 Terrestrial Ecology Unit, Department of Biology, Faculty of Sciences, Ghent University, Ghent, Belgium, 4 School of Biological Sciences, University of Utah, Salt Lake City, Utah, United States of America, 5 Institute of Molecular Biology & Biotechnology, Foundation for Research & Technology Hellas, Heraklion, Crete, Greece, 6 Laboratory of Pesticide Science, Department of Crop Science, Agricultural University of Athens, Athens, Greece, 7 Department of Chemistry, University of Crete, Heraklion, Crete, Greece, 8 Henry Eyring Center for Cell and Genome Science, University of Utah, Salt Lake City, Utah, United States of America

☯ These authors contributed equally to this work.
* thomas.vanleeuwen@ugent.be

**Data Availability Statement:** Sequence data has been deposited at the Sequence Read Archive

## Abstract

Chemical control strategies are driving the evolution of pesticide resistance in pest populations. Understanding the genetic mechanisms of these evolutionary processes is of crucial importance to develop sustainable resistance management strategies. The acaricide pyflubumide is one of the most recently developed mitochondrial complex II inhibitors with a new mode of action that specifically targets spider mite pests. In this study, we characterize the molecular basis of pyflubumide resistance in a highly resistant population of the spider mite *Tetranychus urticae*. Classical genetic crosses indicated that pyflubumide resistance was incompletely recessive and controlled by more than one gene. To identify resistance loci, we crossed the resistant population to a highly susceptible *T. urticae* inbred strain and propagated resulting populations with and without pyflubumide exposure for multiple generations in an experimental evolution set-up. High-resolution genetic mapping by a bulked segregant analysis approach led to the identification of three quantitative trait loci (QTL) linked to pyflubumide resistance. Two QTLs were found on the first chromosome and centered on the cytochrome P450 *CYP392A16* and a cluster of *CYP392E6-8* genes. Comparative transcriptomics revealed a consistent overexpression of *CYP392A16* and *CYP392E8* in the experimental populations that were selected for pyflubumide resistance. We further corroborated the involvement of CYP392A16 in resistance by *in vitro* functional expression and metabolism studies. Collectively, these experiments uncovered that CYP392A16 N-demethylates the toxic carboxamide form of pyflubumide to a non-toxic compound. A third QTL coincided with cytochrome P450 reductase (*CPR*), a vital component of cytochrome P450 metabolism. We show here that the resistant population harbors three gene copies of *CPR* and that

(PRJNA596790). Phenotypic quantification data is available at S1 Data.

**Funding:** This work was supported by the Research Foundation - Flanders (FWO) [grant G009312N and grant G053815N to T.V.L.] and the Research Council (ERC) under the European Union's Horizon 2020 research and innovation program [grant 772026-POLYADAPT to T.V.L. and 773902–SUPERPEST to J.V. and T.V.L.], and by the USA National Science Foundation (award 1457346 to R.M.C). N.W. was supported by a Research Foundation - Flanders (FWO) and a BOF (Ghent University) post-doctoral fellowship (12T9818N and 01P03420, respectively). The funders had no role in study design, data collection and analysis, decision to publish, or preparation of the manuscript.

**Competing interests:** The authors have declared that no competing interests exist.

this copy number variation is associated with higher mRNA abundance. Together, we provide evidence for detoxification of pyflubumide by cytochrome P450s that is likely synergized by gene amplification of *CPR*.

## Author summary

Our understanding of the causal genetic variants that drive the evolution of quantitative traits, such as polygenic pesticide resistance, remains very limited. Here, we followed a high-resolution genetic mapping approach to localize the genetic variants that cause pyflubumide resistance in the two-spotted spider mite *Tetranychus urticae*. Three well-supported QTL were uncovered and pointed towards a major role for cytochrome P450-mediated detoxification. *Cis*-regulatory variation for cytochrome P450s was observed, and *in vitro* cytochrome P450 experiments showed that pyflubumide was metabolized into a non-toxic derivate. A third QTL centered on cytochrome P450 reductase (*CPR*), which is required for cytochrome P450 activity, and is amplified in pyflubumide resistant populations. Our results indicate that pyflubumide resistance is mediated by cytochrome P450 detoxification that is enhanced by gene amplification at the *CPR* locus.

## Introduction

Arthropod herbivores cause over US$450 billion in crop yield loss worldwide [1], and pesticides of various modes of action have been developed to control populations of these pests. Unfortunately, our past and continuing dependence on chemical control strategies is driving the evolution of pesticide resistance in pest populations [2]. Pesticide resistance can be achieved by genetic changes in the molecular target of pesticides (toxicodynamic resistance). These genetic changes are most often non-synonymous substitutions in coding sequences, but (dosage sensitive) gene amplification and differential transcriptional regulation have also been reported, all of which lead to decreased pesticide sensitivity [2,3]. Alternatively, genetic variants that alter the penetration, metabolism, sequestration, efflux, and excretion of pesticides can underlie pesticide resistance (toxicokinetic resistance) [2,4]. Multi-gene families such as cytochrome P450 monooxygenases (CYP), ABC transporters, glutathione S-transferases, and carboxyl/cholinesterases are often associated with the metabolic changes that lead to toxicokinetic resistance. Transcriptomic studies of major global pests have shown that heritable changes in the transcriptional regulation of detoxification genes often strongly associate with pesticide resistance [5–8]. However, our understanding of the causal genetic variation that regulates these transcriptional changes remains limited, and comes disproportionally from insect model species such as drosophilid fruit flies. Genetic changes in both *cis* and *trans* regulation can alter the transcription of detoxification genes [2]. For instance, in *Drosophila melanogaster*, a transposable element insertion in the 5'-UTR region of *Cyp6g1* leads to higher transcript levels that underlie resistance to DDT [9,10]. *Cis*-regulatory variation underpinning the overexpression of cytochrome P450s have been found in a number of insects, including the African malaria vector *Anopheles funestus* [11,12]. Only infrequently, changes in detoxification gene dosage has been associated with resistance, as for resistance to organophosphates by esterase sequestration in insect pests [3,13]. Alternatively, genetic variation affecting the coding sequence of a detoxification gene can also underlie toxicokinetic resistance. A single point mutation in the *GSTe2* gene of *A. funestus* results in pyrethroid and DDT resistance [14]. In dipterans, mutations in the αE7 esterase gene have been found to contribute to a broad

resistance to organophosphates [15,16], whereas in the moth *Helicoverpa armigera*, a gene conversion event generated *CYP337B3* which codes for a novel cytochrome P450 that is able to metabolize fenvalerate [17,18].

The genetic architecture of pesticide resistance can therefore be variable, and can have independent evolutionary origins across different pest populations [19–22]. While monogenic resistance is typically toxicodynamic, and can lead to very high levels of pesticide resistance [22–24], polygenic resistance is nonetheless very common, and has been more challenging to study. For this mode of resistance inheritance, neither the molecular mechanisms that are involved, nor their additive or synergistic effects, are well understood. However, the accelerating development of genetic and genomic resources in diverse arthropods enables high-resolution genetic mapping approaches to characterize complex genetic architectures of pesticide resistance [25,26]. One such approach is bulked segregant analysis (BSA) genetic mapping [25]. A BSA approach relies on crossing parental strains with contrasting phenotypes, bulking (pooling) the segregating progeny that display phenotypic extremes, and identifying genomic intervals with divergent allele frequencies across progeny bulks. BSA genetic mapping has been successfully applied in arthropods to uncover both monogenic and quantitative traits, including pigmentation and host plant use [23,27–33]. The haplodiploid spider mite *Tetranychus urticae* is a major global agricultural pest which rapidly develops resistance to pesticides of various modes of action [34]. As revealed by recent studies, *T. urticae* is highly amenable to BSA genetic mapping, with QTL, including for quantitative pesticide resistance, narrowed to small genomic intervals that have suggested specific candidate genes and causal alleles [29,33]. In some instances, known causal variants were recovered [35–38], confirming the validity of the narrow genomic intervals identified by BSA studies using large, highly replicated *T. urticae* experimental populations [25].

The development of pesticides with novel modes of action is crucial to maintain control over pest populations that have developed multi-resistance. Pyflubumide is a carboxanilide acaricide and inhibits mitochondrial complex II, or succinate dehydrogenase. This recently developed compound is highly selective to spider mite pest species [39,40] and acts as a pro-acaricide that requires bioactivation into a toxic derivative. Pyflubumide is converted within the spider mite body into its active deacylated metabolite (carboxamide form, NNI-0711-NH) that strongly inhibits the mitochondrial complex II through binding to the quinone-binding pocket [39,40]. In the current study, we characterize the genetic basis of resistance to this novel acaricide using two resistant *T. urticae* strains (JPR-R1 and JPR-R2). Target-site resistance is not involved in high-level pyflubumide resistance in these strains, and synergism/antagonism bioassays strongly suggested the involvement of cytochrome P450s [41]. In this study, we employed high-resolution BSA genetic mapping and uncovered three QTL associated with pyflubumide resistance. In parallel, transcriptomic analyses were conducted on these experimental populations to further characterize the molecular mechanisms of pyflubumide resistance. *In vitro* functional characterization of the cytochrome P450 *CYP392A16*, a QTL candidate, suggested a role for resistance to pyflubumide and its active (deacylated) carboxamide metabolite. Additionally, another QTL interval was centered on cytochrome P450 reductase (*CPR*), which is required for cytochrome P450 activity, and is amplified in the resistant populations. Together, our results indicate that the mechanism of pyflubumide detoxification involves cytochrome P450 activity enhanced by gene amplification at the *CPR* locus.

## Results

### Mode of inheritance of pyflubumide resistance

Previously, we selected a field-collected strain for higher resistance levels to pyflubumide and obtained two highly resistant strains, JPR-R1 and JPR-R2 [41]. In this study, we first confirmed

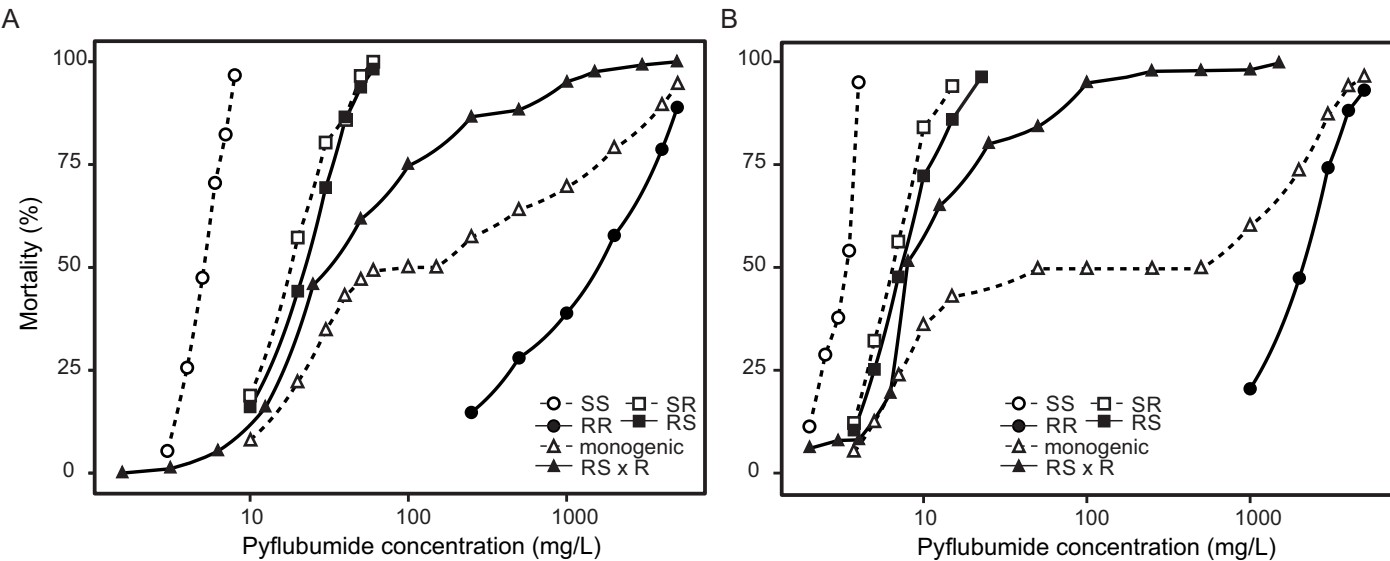

**Fig 1. Pyflubumide resistance has a polygenic basis in JPR-R1 and JPR-R2.** (A) Dose-response relationships of pyflubumide toxicity on JPR-R1, JP-S (circles), reciprocal crosses (squares), and back-crosses (triangles). (B) Dose-response relationships of pyflubumide toxicity on JPR-R2, Wasatch (circles), reciprocal crosses (squares), and back-crosses (triangles). Panels show that pyflubumide resistance has an incomplete recessive mode of inheritance in both resistant strains. After back-crossing, the observed dose-response relationships significantly differ from those calculated under the hypothesis of a recessive monogenic mode of inheritance (p< 0.001) (black versus white triangles, respectively), indicating that pyflubumide resistance is determined by multiple loci in both JPR-R1 and JPR-R2.

the high levels of pyflubumide resistance and uncovered $LC_{50}$ values higher than 1,000 mg/L of pyflubumide for both strains (S1 Table). Reciprocal crosses between resistant (JPR-R1 and JPR-R2) and susceptible (JP-S and Wasatch) strains revealed that pyflubumide resistance is not maternally inherited and has an incomplete recessive mode of inheritance (Fig 1 and S1 Table). The dose-response relationships of the two back-crossed populations were significantly different from the expected curves for a recessive monogenic mode of inheritance (p< 0.001) [42]. These results indicate that pyflubumide resistance in both JPR-R1 and JPR-R2 was determined by multiple loci.

## Response to pyflubumide selection pressure in an experimental evolution set-up

To further characterize the genetic architecture of pyflubumide resistance, we crossed a single haploid JPR-R1 male to diploid females of the susceptible inbred WasX strain. Mortality at 10, 50, 100, and 500 mg/L of pyflubumide was significantly different between JPR-R1 and WasX ($X^2$ = 41.631, df = 1, p < 0.0001) (Fig 2). The segregating population resulting from the WasX × JPR-R1 cross was allowed to expand and develop for two months (approximately three-four generations), after which 11 pairs (22 populations) were set-up and allowed to propagate for approximately 35 generations. In this pairwise set-up, one sister population was exposed to pyflubumide on sprayed plants (a selected population), whereas the other was maintained on unsprayed plants (an unselected, control population). After a prolonged selection with increasing doses of pyflubumide, all populations were phenotyped, revealing that mortality at four pyflubumide doses was significantly lower for pyflubumide-selected populations as compared to the control populations (Fig 2) ($X^2$ = 29.4361, df = 1, p<0.0001). Mortality was also significantly different across the four pyflubumide doses (each p-value < 0.0001).

We subsequently sequenced DNA and RNA from the 22 experimental populations, and investigated both genomic and transcriptomic responses to pyflubumide selection by a

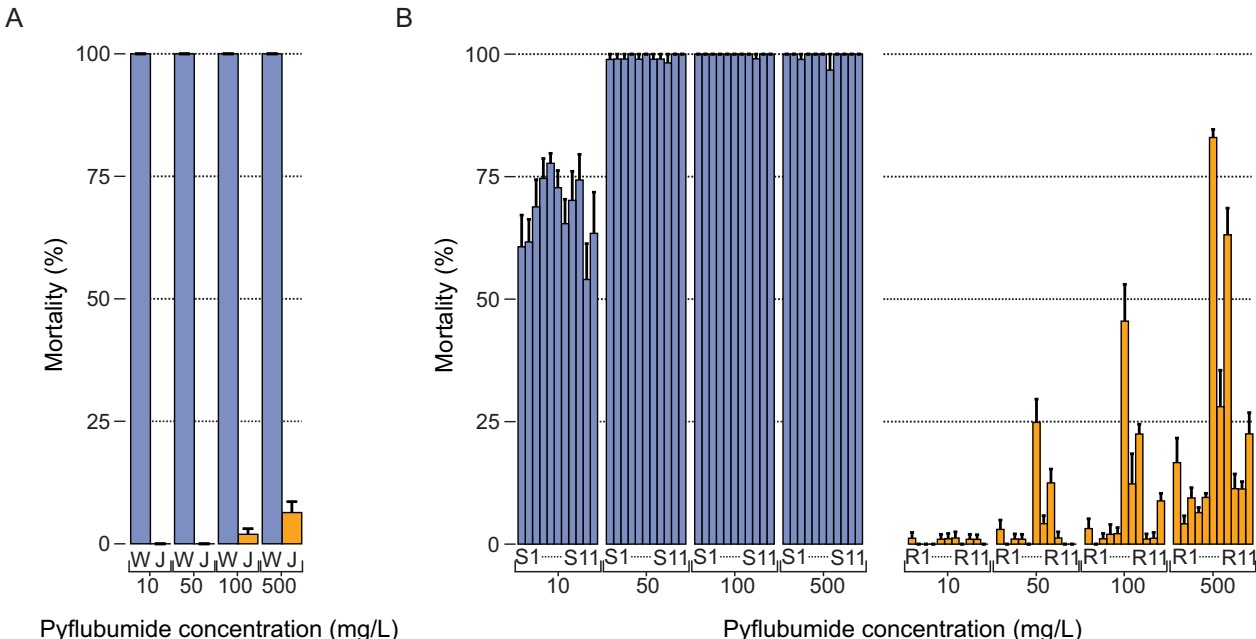

**Fig 2. Selection of segregating populations resulted in high levels of pyflubumide resistance.** (A) Mortality after exposure to 10, 50, 100, and 500 mg/L of pyflubumide of WasX (W) and JPR-R1 (J) (depicted in blue and orange, respectively), the two parental strains of the BSA cross. Error bars represent standard errors. Mortality was significantly different at every pyflubumide dose between JPR-R1 and WasX (p < 0.0001). (B) Mortality after exposure to 10, 50, 100, and 500 mg/L of pyflubumide of control (S1-S11) and pyflubumide-selected (R1-R11) populations (depicted in blue and orange, respectively). Error bars represent standard errors.

principal component analysis (PCA). Using per sample allele frequencies at 214,502 high-quality SNP loci that distinguished JPR-R1 and WasX, control populations clustered closely together and the pyflubumide-selected populations were distinct from these controls along principal component 1 (PC1). Compared to the control populations, pyflubumide-selected populations were more dispersed; most notably, for an unknown reason, but that might have resulted from a population bottleneck that can occur in populations under pesticide selection pressure, population R6 stood out from the other selected populations along principal component 2 (PC2) (Fig 3A). Using our RNAseq data that consisted of a single RNA replicate per experimental population, the genome-wide transcript levels also clearly separated the control and pyflubumide-selected populations along PC1 (Fig 3B). Considering each experimental population as a replicate, a total of 413 genes were identified as differentially expressed between the pyflubumide-selected and control populations (S2 Data).

## Genetic mapping uncovers multiple QTL associated with pyflubumide selection

To locate the genomic regions that underlie pyflubumide resistance in the pyflubumide-selected populations, we employed a bulked segregant analysis (BSA) approach using high-quality SNP loci [25,29]. Genome-wide JPR-R1 allele frequencies between pyflubumide-selected and susceptible (control) populations (Fig 4A) and the matched pairs of populations (Fig 4B) revealed large deviations in allele frequencies on chromosomes 1 and 2. Using previously developed permutation-based methods for QTL detection with replicated paired selected and control populations [29], and with a false discovery rate (FDR) of 0.05, we identified two QTL on chromosome 1 (hereafter QTL-1 and QTL-2) and one on chromosome 2 (hereafter

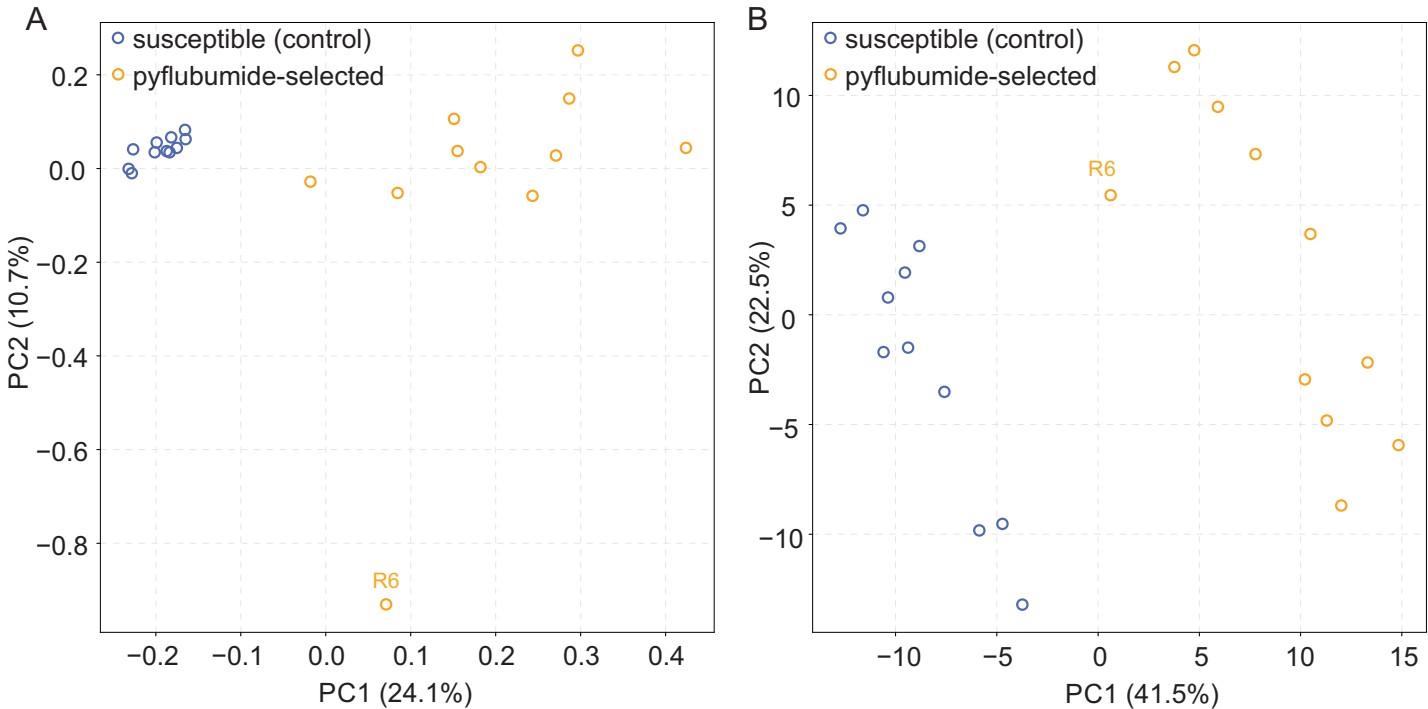

**Fig 3. Selection for pyflubumide resistance is associated with a genomic and transcriptomic response.** (A) A principal component analysis of the eleven susceptible (control) and pyflubumide-selected populations using genome-wide allele frequency data at informative genomic loci. (B) A principal component analysis of the eleven susceptible (control) and pyflubumide-selected populations using genome-wide transcript levels. For both panels, each circle represents an individual population, the R6 population is labelled, treatment is color-coded as indicated (top left), and the percentage of explained variance is given in parentheses for each principal component (PC).

QTL-3) in the pyflubumide-selected populations (Figs 4C and S1). In all cases, alleles from the resistant JPR-R1 parent were selected, and for all pyflubumide-selected populations, near-fixation of the JPR-R1 haplotype was observed at the peaks of the three QTL regions (S1 Fig). Population R6 exhibited the lowest JPR-R1 allele frequency near the peaks of QTL-2 and QTL-3 (S1 Fig), a finding that is consistent with its lower resistance levels (Fig 2). We also noted two additional genomic regions where the allele frequency differences nearly exceeded the significance threshold (hereafter referred to as subsidiary peaks 1 and 2) (Fig 4C). Finally, a BSA mapping approach was also performed using informative SNP loci identified via the RNAseq data. This analysis further confirmed the three QTL, as well as the two subsidiary peaks (S2 Fig). Of the 413 differentially expressed genes (DEGs) detected between the selected and unselected populations, 123 (30.8%) were within the three top QTL genomic intervals defined as the regions with allele frequencies similar to those of the peaks (at least one of five consecutive windows having allele frequency values within 0.005 of the peak values). Specifically, 24 DEGs were located within the top 180 kb QTL-1 region, whereas 25 and 74 DEGs were within the 130 kb QTL-2 and 110 kb QTL-3 genomic intervals, respectively. The linkage information suggests that *cis*-regulation could play a dominant role in the different expression levels of these 123 DEGs (note that in the three QTL regions there are large differences in the frequency of the haplotypes from the two parents between the selected and unselected replicates, so *cis*-expression QTLs (eQTLs) are expected to be uncovered compared to distant genomic locations where haplotype frequencies are not significantly different). The RNA levels of the other (QTL-interval distant) 290 DEGs are presumably *trans*-regulated, with the different RNA levels controlled by genetic variants within the three QTL regions that genetically differentiate the selected and control populations.

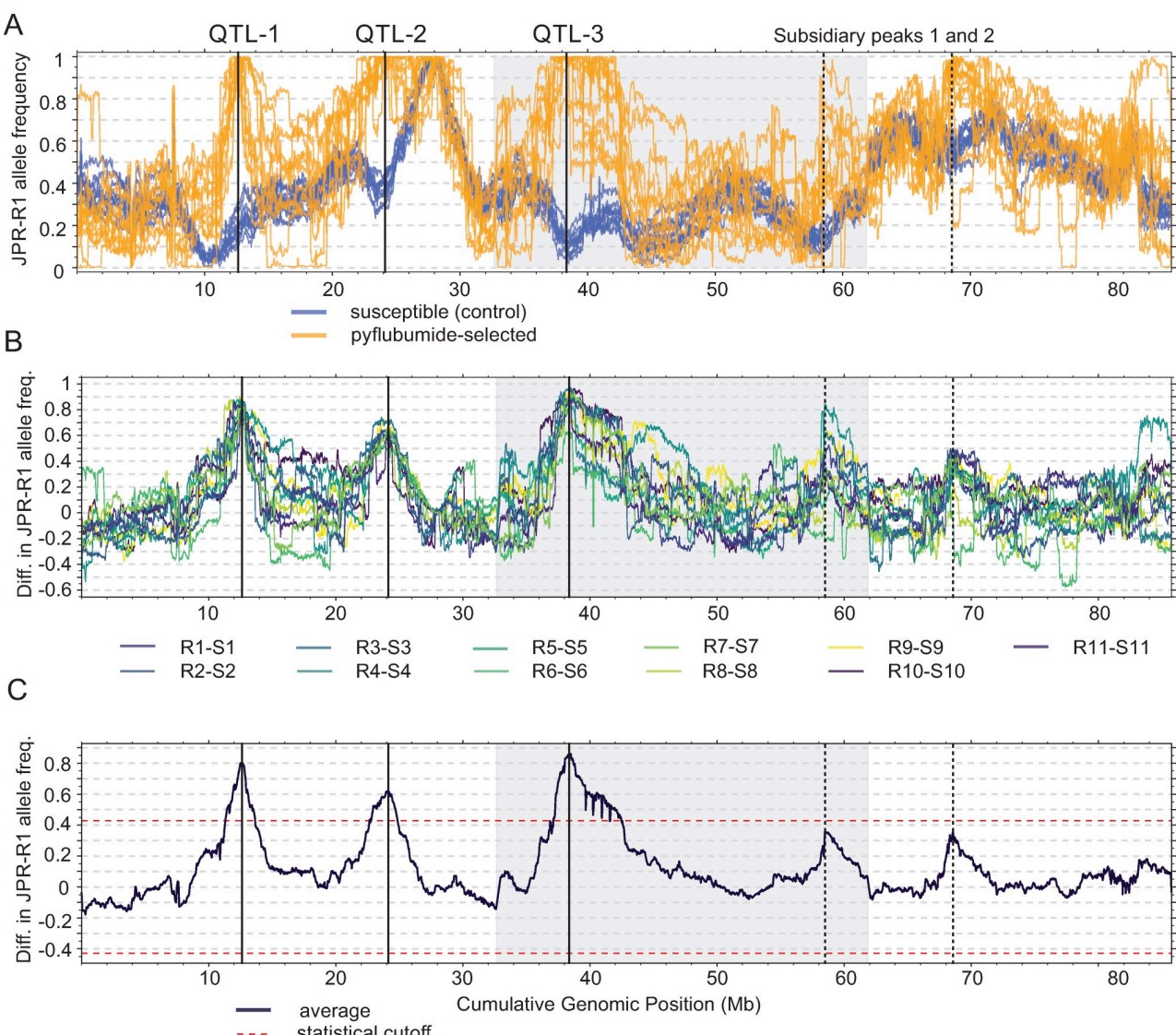

**Fig 4. BSA genetic mapping reveals three QTL for pyflubumide resistance.** (A) Frequency of JPR-R1 alleles in the susceptible control (blue lines) and pyflubumide-selected populations (orange lines) as assessed in a sliding window analysis. (B) Genome-wide differences in JPR-R1 allele frequencies between the eleven paired populations (R1-S1 to R11-S11), as assessed in a sliding window analysis. (C) Averaged genome-wide difference in JPR-R1 allele frequency using the eleven paired populations (R1-S1 to R11-S11). Dashed red lines delineate statistical significance for QTL detection as assessed by a permutation approach (FDR of 5%). In all panels, chromosomes are ordered by decreasing length and are indicated by alternating shading. Three QTL (QTL-1, QTL-2, and QTL-3) exceed the 5% FDR threshold; two sub-threshold peaks are also labeled (subsidiary peaks 1 and 2).

### Candidate genes for pyflubumide resistance are linked to cytochrome P450-mediated detoxification

In *Tetranychus* spider mites, previous BSA studies of similar experimental design revealed that the average of peaks from replicated paired populations are often within several tens of kb from causal genes and variants [25,27,33]. Therefore, we examined ~75 kb genomic intervals that bracket the average of the BSA peaks to detect potential causal candidate genes (Fig 5 and S2 Data). For QTL-1, the cytochrome P450 *CYP392A16* was the only gene in the interval with a predicted metabolic function that could be readily linked with xenobiotic metabolism of

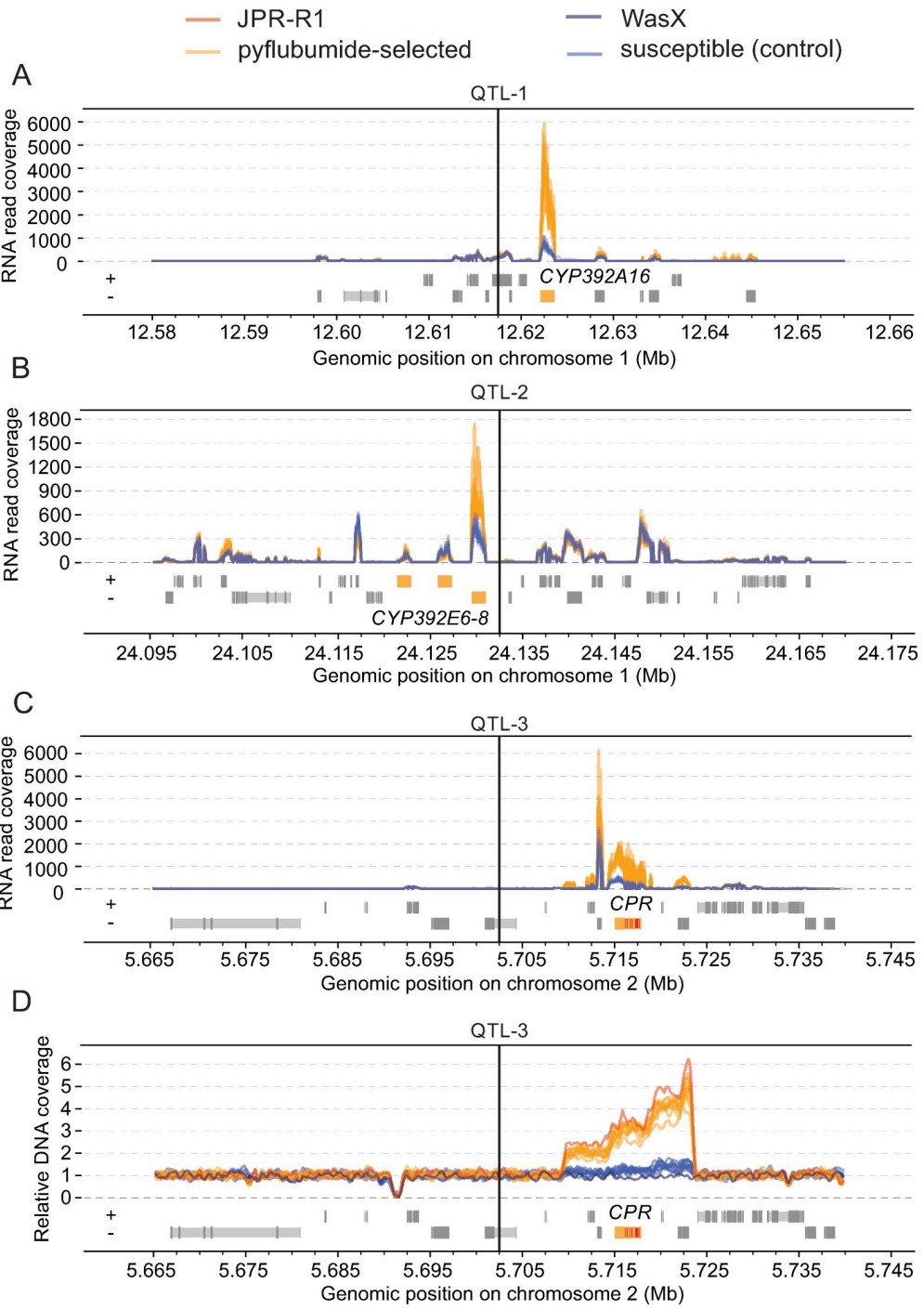

**Fig 5. Genes at significant BSA peaks suggest a prominent role for cytochrome P450 activity in resistance to pyflubumide.** RNAseq read coverage and gene models in ~75kb genomic windows surrounding QTL-1 (A), QTL-2 (B), and QTL-3 (C). (D) Relative DNA coverage and gene models in a ~75kb genomic window surrounding QTL-3. For all panels, vertical lines indicate the locations of the averaged BSA peaks on the first (A and B) and second chromosome (C and D). Candidate genes *CYP392A16* (A), *CYP392E6-8* (B), and *CPR* (C and D) are highlighted in yellow and red (exons and introns, respectively). Otherwise, coding exons and introns are depicted as dark gray and lighter gray boxes, respectively. Symbols + and–denote forward and reverse gene orientations. Coverage is color-coded according to treatment and strain (legend, top).

arthropods. *CYP392A16* was within 10 kb of the averaged BSA peak (Fig 5A), and was one of three genes within the ~75 kb genomic interval with significantly higher expression in pyflubumide-selected populations compared to controls (S2 Data), a potential signal of *cis*-regulatory variation. Specifically, transcript levels of *CYP392A16* were ~five-fold higher in pyflubumide-selected populations (Fig 5A, S3 Fig and S2 Data). DNA read coverage indicated that the segregating populations and the two parental strains had the same copy number of *CYP392A16* (a single copy), ruling out copy number variation (CNV) as underpinning the higher transcript levels of *CYP392A16* (S4 Fig). The ~75 kb genomic interval of QTL-1 also included *tetur06g04470*, a gene that codes for a protein that is classified as a member of the AIP/AIPL1 family (IPR039663), a group of proteins that include aryl hydrocarbon receptor (AhR)-interacting proteins. AhR participates in a signaling pathway that induces the transcription of detoxification genes in insects and changes in this pathway can promote pesticide resistance [43,44]. However, no significant differential transcription, nor non-synonymous single-nucleotide differences were observed for *tetur06g04470*, suggesting that *tetur06g04470* likely does not contribute to pyflubumide resistance in our strains.

A cluster of three cytochrome P450s, *CYP392E6-8*, was located very close to the BSA peak of QTL-2 (Fig 5B), and no other genes with obvious roles in arthropod detoxification were located in the interval. *CYP392E8*, which is located only 2 kb from the QTL-2 peak, was the only gene in the region that exhibited a significant increase in transcription in pyflubumide-selected populations (a fold change of ~2) (Fig 5 and S2 Data). Although substantial CNV for this cytochrome P450 gene cluster has been reported in other *T. urticae* strains [29], all *CYP* genes in the interval were single copy as revealed by genomic read coverage in the parental and segregating populations (S4 Fig).

No cytochrome P450s or genes encoding other detoxification enzymes were apparent at the QTL-3 peak region. However, NADPH cytochrome P450 reductase (*tetur18g03390*, *CPR*), the necessary electron donor for cytochrome P450 metabolism, was within 20 kb of the averaged BSA peak for QTL-3 (Fig 5). Previous work with *T. urticae* revealed an association between a *CPR* coding sequence variant and cytochrome P450-mediated detoxification of pesticides [29,33]; however, short-read DNA alignments revealed that *CPR* harbored no coding sequence differences between JPR-R1 and WasX. However, DNA read coverage revealed an ~14 kb genomic region that is copy number variable between the parental strains, and that harbors five genes including *CPR* (Fig 5D and S2 Data). Upstream of *CPR*, two genes coding for a hypothetical protein without clear homology to functionally characterized protein domains (*tetur18g03400*) and a protein that harbors a folate-binding domain (*tetur18g03410*, PF01571 —cl33847) were within the copy number variable region of JPR-R1. Downstream of *CPR*, two genes coding for a ribosomal protein (*tetur18g03380*, PF00238—PTZ00054) and a archease protein (*tetur18g03370*, PF01951—MTH1598/TM1083) were also copy number variable. The short-read alignments revealed no variable sites within the copy number variable genomic region of JPR-R1, indicating that the different copies of this region are identical. Whereas WasX and the control populations were predicted to have a single copy of the region harboring *CPR*, for JPR-R1 and the resistant populations variable gene copies were predicted, with JPR-R1 having approximately three copies of *CPR*. For *CPR*, an additional copy number assay using an independent method (quantitative PCR) confirmed that *CPR* was present in an ~3:1 ratio in the genomes of pyflubumide-selected versus control populations (S3 Fig). This ratio was mirrored in the RNAseq read data where up-regulation of *CPR* transcription was ~3.66 in pyflubumide-selected versus control populations (Figs 5 and S3). Collectively, this indicates that CNV of *CPR* resulted in higher mRNA levels in JPR-R1 and pyflubumide-selected populations. It was also striking that in the control (unselected) populations, the JPR-R1 allele frequencies dipped to near zero coincident with the *CPR*-associated QTL-3 peak (Figs 4A and

S1), potentially reflecting a negative pleiotropic effect on fitness that becomes apparent in absence of pyflubumide selection (see Discussion). This pattern was absent or less apparent for the other QTL peaks (for instance, a superficially similar pattern for QTL-1 is likely explained by linkage drag from a proximal locus at ~10 Mb on chromosome 1 that is unrelated to pyflubumide selection).

Consistent with previous work [41], our BSA genetic mapping with JPR-R1 excluded target-site insensitivity as a contributing mechanism to pyflubumide resistance. Only one of the genes encoding a mitochondrial complex II subunit, the known molecular target(s) of pyflubumide, was near a QTL peak (*tetur08g03210*, which is located ~640 kb from the BSA peak for QTL-2). Although the two subsidiary BSA peaks did not reach the significance threshold for QTL detection (Fig 4C), we nonetheless analyzed their genic content. For subsidiary peak 1, no obvious candidate detoxification genes were found within the peak ~75 kb region, nor were any genes differentially expressed. For subsidiary peak 2, two *CYP* genes, *CYP392B2* and a predicted *CYP* pseudogene (*tetur02g06640*) in the London reference strain, were within the peak ~75 kb region, with the latter exhibiting lower RNA abundance in the pyflubumide-selected populations (a fold change of -2) (S2 Data). Several non-synonymous single-nucleotide differences were observed in both *CYP* genes between JPR-R1 and WasX. Despite the lack of statistical significance for subsidiary peak 2, *CYP392B2* is of interest as it was previously associated with pyflubumide resistance [45] (and see Discussion).

## CYP392A16 demethylates the toxic pyflubumide carboxamide

To test if *CYP392A16* contributes to pyflubumide resistance in JPR-R1, we functionally expressed the gene to determine if the recombinant protein was active against pyflubumide and/or its toxic carboxamide metabolite (NNI-0711-NH) (Fig 6). The recombinant protein was well-folded (S5 Fig), exhibited activity towards the model substrate L-ME EGE (4 +/- 0.5 SE pmol D Luciferin/min/pmol P450), and membrane preparations exhibited CPR activity (2407 +/- 33.8 nmol/min/mg). By HPLC, substrate depletion was not observed when recombinant CYP392A16 was incubated with the proacaricide pyflubumide (Fig 6). In contrast, an NADPH-dependent depletion of its active, toxic NNI-0711-NH derivative was observed, along with the parallel formation of an unknown compound (eluting at 8.5 min and 6.7 min, respectively), after incubation with recombinant CYP392A16. Chromatograms showed no change in the amount of the carboxamide when incubations were performed in the absence of an NADPH regenerating system, which is necessary for cytochrome P450 activity (Fig 6). The ability of CYP392A16 to metabolize the pyflubumide carboxamide was further characterized by measuring substrate-dependent reaction rates. Its depletion rate as a function of substrate concentration revealed Michaelis-Menten kinetics ($R^2$ of fitted curve = 0.96, $V_{max}$ = 17.42 (± 1.07 SE) pmol of depleted deacylated pyflubumide metabolite/min and $K_m$ = 17.79 (± 3.29 SE) μM) (Fig 6).

CYP392A16 metabolized 21% of the pyflubumide carboxamide NNI-0711-NH (with a concentration of 25 μM) after a 1-hour incubation. HPLC-MS/MS analysis of the reaction mixtures pointed to N-demethylation as the likely mechanism of the CYP392A16 reaction (Fig 7) [46]. Demethylation is expected to result in a mass difference of -14 Da between the NNI-0711-NH and reaction metabolite product ions. This was reflected in almost all substrate and reaction product ions formed by collision-induced dissociation (Fig 7). More specifically, the product ion MS data indicated that demethylation involves a pyrazole methyl group. We arrived at this conclusion because pyrazole-containing product ions originating from the reaction product are 14 Da lower than their corresponding substrate product ions of NNI-0711-NH (Fig 7).

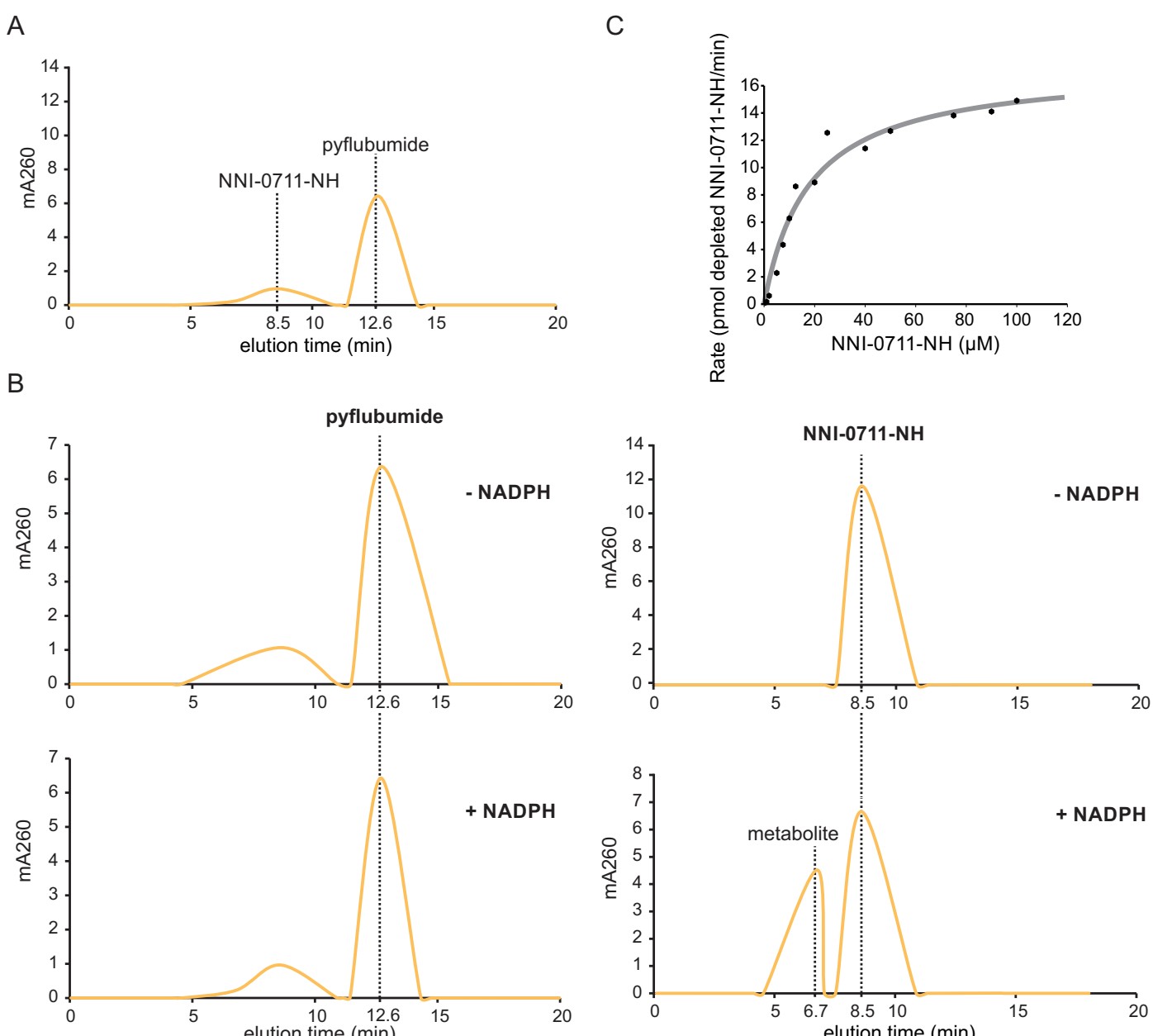

**Fig 6. CYP392A16 is active towards the toxic deacylated pyflubumide metabolite, NNI-0711-NH, but not pyflubumide.** (A) HPLC chromatogram of pyflubumide and its toxic metabolite NNI-0711-NH, eluting at 12.6 and 8.5 min, respectively. (B) HPLC chromatograms show a NADPH-dependent depletion of the toxic NNI-0711-NH (eluting at 8.5 min) and the corresponding formation of a reaction product (eluting at 6.7 min). In contrast, reactions carried out in the absence and presence of a NADPH-regenerating system showed no change in the chromatogram profile with pyflubumide eluting at 12.6 min. Absorbance curves are plotted in yellow (A and B). (C) Michaelis–Menten kinetics of the depletion of NNI-0711-NH by CYP392A16. Plotted values represent the mean. The curve was calculated by non-linear regression.

## Discussion

Pesticide resistance is a rapidly evolving trait within the arthropod phylum [19,47]. Despite its economic importance and its potential to contribute to our understanding of general evolutionary processes, several questions about the genetic architectures underlying pesticide resistance are still outstanding, especially in the case of polygenic resistance. Pyflubumide is a

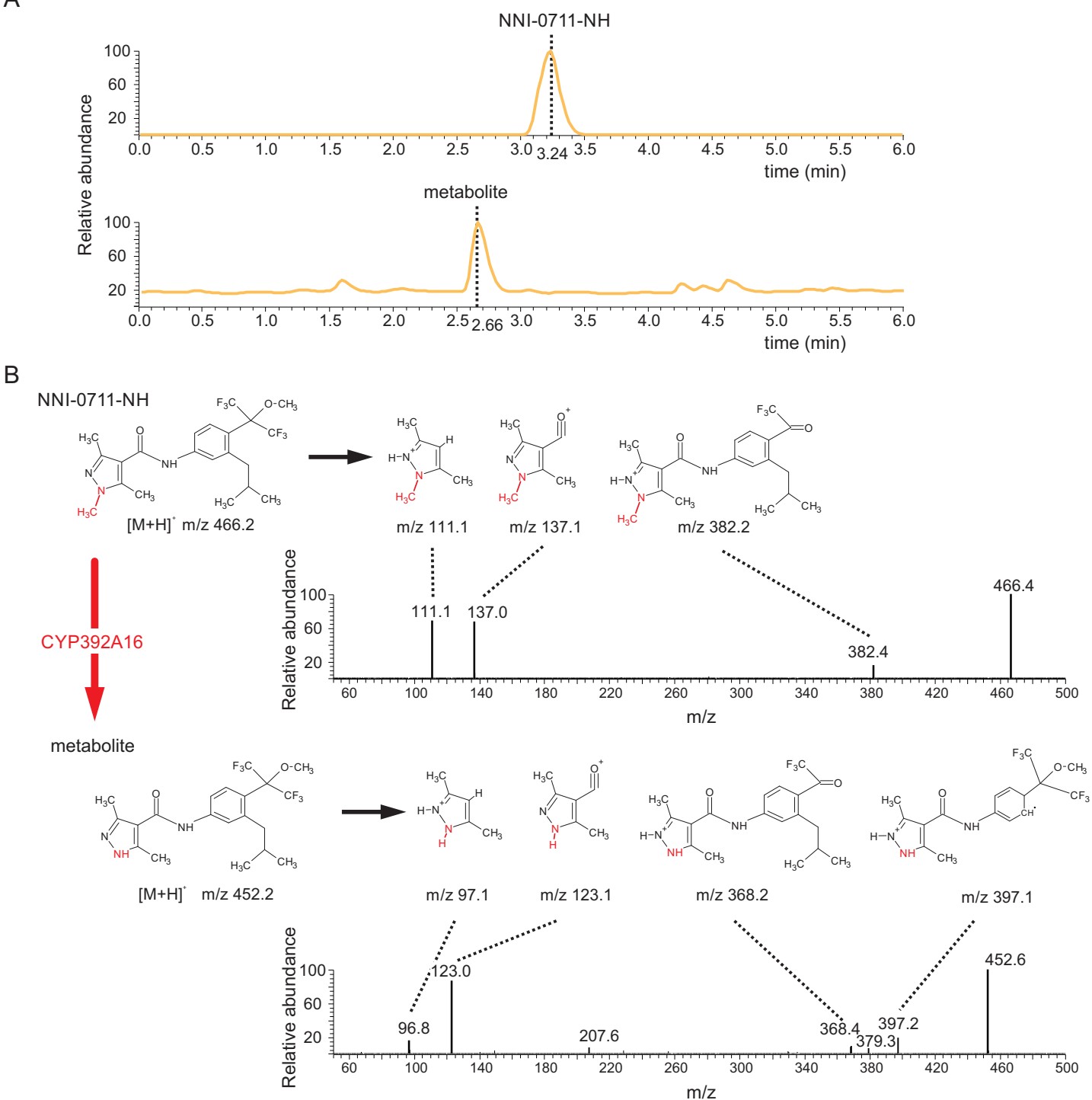

**Fig 7. The toxic pyflubumide metabolite NNI-0711-NH is demethylated by CYP392A16.** (A) Total ion chromatograms and product ion mass spectra obtained from the collision induced dissociation of NNI-0711-NH and the reaction metabolite molecular ions (m/z 466 and m/z 452 respectively) as obtained following their HPLC separation. TIC values for NNI-0711-NH and the reaction metabolite were 5.01E5 and 1.07E4, respectively. (B) Proposed product ions for NNI-0711-NH (m/z 466) and the reaction metabolite (m/z 452). Demethylation, which is expected to proceed via hydroxylation, results in a mass difference of -14 Da between NNI-0711-NH and the metabolite. The proposed demethylation site is indicated by red font. This scenario was confirmed in almost all product ions formed by collision induced dissociation.

recently developed carboxanilide acaricide that is highly effective against plant-feeding spider mites [39–41,48]. Based on synergism/antagonism assays of previous work [41], the resistant *T. urticae* JPR-R1 and JPR-R2 strains most likely rely on cytochrome P450-mediated detoxification to overcome pyflubumide toxicity. The two strains also exhibit highly similar constitutive and plastic transcriptional patterns on a genome-wide level [41]. Here, we uncovered that pyflubumide resistance of both JPR-R1 and JPR-R2 is determined by multiple loci. Considering the two strains share a common ancestral genetic background with already moderate levels of pyflubumide resistance [41], these findings further suggest that JPR-R1 and JPR-R2 likely share (some) factors underlying pyflubumide resistance. Using JPR-R1 and susceptible WasX as parents, BSA genetic mapping revealed three major QTL that are associated with pyflubumide resistance based on the segregating genomes of 11 pyflubumide-selected and control population pairs. Interestingly, a BSA genetic mapping approach based on RNAseq data (instead of genomic DNA) uncovered highly similar allele frequency differences, mirroring the QTL pattern obtained with DNA. Less biological tissue is needed to obtain sufficient RNA, as compared to DNA, for high-quality sequencing, and our findings highlight the feasibility of using RNAseq data for BSA genetic mapping, a consideration that is relevant for microarthropods for which biological material is often limiting.

*CYP392A16* was located at the center of QTL-1, was over-expressed in pyflubumide-selected populations and the JPR-R1 parent, and metabolized the toxic pyflubumide carboxamide *in vitro*. Incubation assays with membrane fractions of recombinant protein further uncovered that CYP392A16 demethylates the active carboxamide into a derivative that is not toxic to spider mites [48], confirming that CYP392A16 catalyzes a detoxification reaction. The *CYP* gene cluster of *CYP392E6-8* was close to the peak of QTL-2 and *CYP392E8* exhibited higher transcription in pyflubumide-selected populations and JPR-R1. In an earlier study, BSA genetic mapping uncovered a broad QTL (referred to as spiro-QTL 2) that included the *CYP392E6-8* gene cluster when replicated populations were selected for resistance to spirodiclofen, a pesticide that targets lipid synthesis [29]. This raises the question of whether the same or a different *CYP* gene within this cluster was the target of selection and whether other *CYP* genes outside this cluster (partially) underpin pyflubumide and spirodiclofen resistance. The high susceptibility of the *T. urticae* SR-VP strain to pyflubumide, the spirodiclofen-resistant parental strain of the earlier BSA study, suggest that different *CYP* genes (or alleles) determine spirodiclofen and pyflubumide resistance [29,41]. *CYP392E9-10* lie ~312 kb away from the *CYP392E6-8* gene cluster but are still within the wider QTL-2 interval. *CYP392E7* and *CYP392E10* are overexpressed in spirodiclofen resistant strains, and previous work was able to show that *CYP392E10* metabolizes spirodiclofen by hydroxylation [37,49]. Using the annotated genome of the London reference strain, QTL-2 is predicted to additionally hold several *CYP* gene fragments and pseudogenes [29,50]. As the *CYP* multi-gene family is characterized by highly dynamic birth and death rates [51,52], it is possible that these genes could be functional in JPR-R1. Further work is needed to fully characterize the functionality and enzymatic abilities of the multiple cytochrome P450s within QTL-2 and to understand why this genomic region has been a locus of selection for both spirodiclofen and pyflubumide resistance.

The causal genetic variants underpinning the over-expression of detoxification genes in resistant populations of agricultural pests has only been characterized in a limited number of cases [2,53–59]. We ruled out CNV of the coding sequences as causal for the over-expression of *CYP392A16* and *CYP392E8* in the pyflubumide resistant populations. Given the QTL peak locations, *cis*-regulatory variation appears to be at least partially responsible for the higher transcript levels. In insects, transcription of detoxification genes is controlled by a number of regulatory pathways, including the AhR/ARNT and CncC/Keap1 cascades, in which *cis*-acting transcription factor binding sites have been identified [2,43]. In contrast, transcriptional

regulation remains poorly understood in mites, and promotor/enhancer analyses are currently not trivial to conduct or interpret. Future studies are therefore needed to test our hypothesis of *cis*-regulation, and to identify the potential *cis*-acting elements and their transcription factor binding partners, using suitable experimental set-ups [60,61].

QTL-3 centered on *CPR*, a necessary redox partner of cytochrome P450-mediated detoxification. Our analyses provide strong evidence for CNV at the *CPR* locus. Pyflubumide resistant populations were estimated to harbor three *CPR* copies by two methods, whereas susceptible populations have a single *CPR* copy. Neighboring genes on both sides of *CPR* also exhibit clear signs of CNV in JPR-R1, but with different copy numbers. The absence of CNV at this locus in other previously characterized *T. urticae* strains and the absence of genetic variation between the different genomic copies of JPR-R1 suggest a recent origin of CNV at the *CPR* locus. The JPR-R1 allele frequencies in the experimental control populations indicate that this copy number variable region could induce pleiotropic fitness penalties that lead to purging in the absence of strong selection pressure (i.e., continuous pyflubumide exposure). Potential fitness penalties may explain why CNV at the *CPR* locus has not yet been associated with other *T. urticae* resistance phenotypes that rely on cytochrome P450-mediated detoxification. The transcriptomic comparisons revealed that relative transcript levels were ~three-fold higher for *CPR* in the pyflubumide-selected populations compared to the controls. The nearly identical DNA and RNA coverage ratios strongly suggest that CNV determines the RNA levels of *CPR* in JPR-R1 by positive dosage effects. Previous studies have linked dosage-sensitive amplification of *CYP* genes to increased detoxification, and in turn, pesticide resistance [56–59]. We hypothesize that dosage-sensitive gene amplification of *CPR* in JPR-R1 strengthens cytochrome P450-mediated detoxification because P450 activity is dependent on the concentration of the CPR-P450 complex [62]. Indeed, reducing *CPR* transcription in *T. urticae* and other arthropod pests by RNA interference is known to increase susceptibility to pesticides [63,64]. Moreover, the combined up-regulation of *CYP392A16*, *CYP392E8* and *CPR* could have additive or synergistic effects, resulting in higher levels of pyflubumide resistance. However, as the relationship between CNV, RNA levels, translation, and phenotypic traits can be complex [65], further work is needed to precisely quantify the impact of the dosage-sensitive CNV at the *CPR* locus on pyflubumide resistance. Using a BSA genetic mapping approach, *CPR* was also found at the center of a QTL associated with both spirodiclofen and Mitochondrial Electron Transport Inhibitors complex I (METI-I) resistance in *T. urticae* [29,33]. In both studies, the non-synonymous substitution D384Y was only observed in the pesticide-selected populations and the resistant parental strains SR-VP and MR-VP. Because METI-I and spirodiclofen resistance is mostly mediated by cytochrome P450s [36,37,66], it was hypothesized that D384Y might strengthen cytochrome P450 detoxification. Here, the coding sequences of the different *CPR* copies were identical and did not display D384Y or any other non-synonymous substitutions, compared to the genome of the reference London strain [29,50].

Previous BSA studies in *T. urticae* uncovered a genetic architecture that involves both toxicodynamic and toxicokinetic resistance to spirodiclofen and METI-Is [29,33]. QTL underlying spirodiclofen and METI-I resistance correspond with the molecular targets acetyl-CoA carboxylase and NADH:ubiquinone oxidoreductase, respectively [29,33,38]. None of the three QTLs of pyflubumide resistance were centered on the target of pyflubumide, strongly indicating that only toxicokinetic mechanisms were involved. Previously, Sugimoto *et al.* identified two QTLs that are associated with pyflubumide resistance in *T. urticae* using a microsatellite linkage map [45]. One QTL is located on the first *T. urticae* chromosome and centers on *tetur01g15710*, the gene that codes for succinate dehydrogenase subunit b (*SdhB*), the molecular target of pyflubumide [39,40]. Genotyping a panel of *T. urticae* strains revealed a strict association between the non-synonymous substitution I260V of *SdhB* and pyflubumide resistance.

The I260V substitution is located at the I269 position of the fungus *Zymoseptoria tritici* (synonym *Mycosphaerella graminicola*) protein which impacts fungicide binding to succinate dehydrogenase and results in resistance in *Z. tritici* [67]. Our read data of the segregating populations did not reveal the I260V variant or other candidate target-site resistance mutations in *SdhB*. Additionally, Sugimoto *et al.* observed that their second broad QTL of ~4.1 Mb on the third *T. urticae* chromosome included three cytochrome P450 genes (*CYP392A8*, *CYP392B1*, and *CYP392B2*) [45]. These *CYP* genes were not differentially expressed between the pyflubumide-selected and control populations and were not included in any of the three well-supported QTL of this study. However, subsidiary peak 2 of the current study largely coincided with this QTL (its peak region also included *CYP392B2*). This could indicate that genetic variation within this genomic interval could have a minor effect on pyflubumide resistance in the segregating populations used in our study. Collectively, however, the two complementary genetic mapping approaches strongly suggest that pyflubumide resistance has been obtained by largely different molecular mechanisms between the strains of the current study and Sugimoto *et al.* Our work therefore adds to a growing body of evidence that populations of pest species can evolve different resistance mechanisms despite the highly specific selective pressures imposed by pesticides [19].

In summary, our work uncovered a complex genetic basis for pyflubumide resistance in the arthropod pest *T. urticae* that likely involves multiple genes with roles in cytochrome P450-mediated metabolism. Our *in vitro* findings suggest that CYP392A16 detoxifies the active form of the proacaricide pyflubumide, and that *cis*-regulatory variation at this gene contributes prominently to pyflubumide resistance. We also uncovered a potential novel molecular mechanism of toxicokinetic resistance–gene amplification of *CPR*–that may act in a synergistic manner.

## Materials and methods

### Mite strains and rearing

Five *T. urticae* strains were used for the genetic crosses: JP-S [68,69], Wasatch [28], WasX, JPR-R1, and JPR-R2. The JPR-R1 and JPR-R2 strains were selected for high levels of pyflubumide resistance from the JP-R strain with two different laboratory selection regimes. JPR-R1 was first selected by spraying pyflubumide on detached mite-infested bean leaves for two generations. Subsequently, both JPR-R1 and JPR-R2 were selected by feeding on pyflubumide sprayed potted bean plants (100 mg/L of pyflubumide) [41]. WasX was derived from the highly inbred Wasatch strain by two sequential rounds of mother-son mating. Unless otherwise stated, all *T. urticae* populations were kept under laboratory conditions (25˚C, 65% relative humidity, and 16:8 hr light:dark photoperiod) and maintained on *Phaseolus vulgaris* cv. 'Speedy'. JPR-R1 and JPR-R2 were kept on plants sprayed with 100 mg/L pyflubumide (commercial formulation, 20% SC (Danikong, Japan), kindly provided by Ralf Nauen).

### Mode of inheritance of pyflubumide resistance

The mode of inheritance of pyflubumide resistance was determined by performing reciprocal crosses between JPR-R1 and JP-S, and JPR-R2 and Wasatch. For each cross, 80 teliochrysalid (virgin) females and 100 adult males were placed on leaf discs and allowed to mate. Females were transferred to new leaf discs every 2 days throughout their oviposition period. During development, 80 F1 teliochrysalid females were isolated for backcrosses to obtain F2 females. Dose-response bioassays were conducted for the parents and F1 and F2 females as previously described [41]. Briefly, 9 cm$^2$ bean leaf discs were sprayed with 1 ml of acaricide solution at 1 bar pressure in a Potter spray tower (deionized water was used as the control). Depending on

the population, 5–13 pyflubumide concentrations were tested using four replicates (20–35 females per replicate). Lethal concentrations and 95% confidence limits were determined using Probit analysis (POLO; LeORa Software). The degree of dominance (D) was calculated using Stone's formula [70]. The expected dose-response relationships for F2 females for monogenic recessive resistance was calculated using c = 0.5 W (F1) + 0.5 W (JPR-R strain), where c is the expected response (mortality) at a given concentration and W is the observed response (mortality) of the parents at the respective concentration [42]. The hypothesis of a monogenic mode of inheritance was tested with $\chi^2$ goodness-of-fit tests.

## Experimental evolution of pyflubumide resistance

A total of 26 teliochrysalid females of JPR-R1 were placed on individual leaf discs. Upon eclosion, unfertilized females were allowed to lay eggs for 2 days and were transferred to a 9 cm$^2$ bean leaf disc. Resistance to pyflubumide was assessed by spraying 4,000 mg/L pyflubumide (1 bar pressure in a Potter spray tower with 2 mg aqueous deposit per cm$^2$) and scoring mortality after 24 hr. Eggs of one of the resistant JPR-R1 females were allowed to hatch and develop into adult males. A single adult male (an isogenic genome, as males are haploid) was selected from this brood and was offered four female teliochrysalids of WasX on a daily basis. In total, 24 resulting females were allowed to produce an F1 generation of more than 500 female mites. The segregating population was expanded and allowed to propagate at a high population size (several hundreds of mites) for approximately two months. From this large population, we established 11 control populations on unsprayed potted bean plants using 650 founding females per control population. After approximately one month, sister populations were generated from these 11 control populations by transferring approximately 650 females to potted bean plants sprayed with 50 mg/L pyflubumide. Sprayed and unsprayed potted bean plants were offered to the 22 experimental populations every other week. During the experiment, selection pressure of pyflubumide was gradually increased until mites no longer exhibited acaricide-related mortality on beans sprayed until run-off with 100 mg/L of pyflubumide (selection was maintained for approximately a year). Dose-response bioassays were subsequently performed on the two parents and the 22 segregating populations as described above using four concentrations (10, 50, 100, and 500 mg/L pyflubumide). Differences in mortality percentages of the two parents were assessed using a generalized linear model with a binomial distribution. A generalized linear mixed model with a binomial distribution was used to analyze the mortality percentages of the pyflubumide-selected and control populations, with selection regime and pyflubumide dosage as fixed effects, and sister pair as a random effect. Percentage of mortality was the dependent variable in both models. Statistical analyses were performed in R using lme4 [71].

## DNA sequencing and variant detection

Immediately after the dose-response bioassays were finalized, DNA was extracted from 1,000 adult females of the 22 segregating populations using a phenol-chloroform method as previously described [27,33]. Six months after the BSA cross, DNA was collected from the genetically diverse JPR-R1 strain and the inbred WasX strain. DNA samples were sequenced at the Huntsman Cancer Institute at the University of Utah to produce Illumina paired-end reads with lengths of 151 bp with mean insert sizes of 450 bp. DNA reads were aligned to the *T. urticae* three-chromosome assembly [29] using the default options of mem in the Burrows-Wheeler Aligner (BWA) version 0.7.17-r1188 [72]. Duplicate reads were marked using the MarkedDuplicates function of Picard version 2.18.11-SNAPSHOT (http://broadinstitute. github.io/picard/). Left-justified indel realignment was performed using the GATK version

4.0.7.0 [73]. Throughout these steps, SAMtools version 1.9 [74] was used to sort and index the reads in each BAM output. Two joint variant calls were performed using GATK version 4.0.7.0; one that included the two parents and the segregating populations, and one without the parents (the latter joint variant call was only used for the PCA). Each variant call was performed with HaplotypeCaller and then merged into a single gVCF file using the GVCFs function, after which the GenotypeGVCFs function was used to jointly predict SNPs and small indels.

## RNA sequencing and differential expression analysis

Thirty-seven days after DNA was extracted from the segregating populations, the pyflubumide selection treatment was terminated by transferring mites from all experimental populations to new unsprayed bean plants where they were allowed to develop for one generation (eleven days). RNA was extracted from 110 adult females using the RNeasy minikit (Qiagen) for every segregating experimental population. Total RNA was sequenced with the Illumina method at NXTGNT (Ghent, Belgium) to give read lengths of 100 bp. The paired-end RNA reads were aligned to the *T. urticae* three-chromosome assembly [29,50] using STAR version 2.5.3a [75] with the 2-pass mode, the maximum intron size set to 20 kb, and without the use of the GFF annotation. SAMtools version 1.9 [74] was used for sorting and indexing the BAM files. Read counts were obtained with HTSeq version 0.11.2 [76] using the *T. urticae* annotation of Wybouw *et al.* [29]. Analysis of gene expression was performed using DESeq2 version 1.24.0 [77]. Differential expression was identified by applying a 2 and 0.05 cut-off for FC and adjusted p-value, respectively. Coverage from the BAM files for downstream analyses was obtained using the defaults of the pileup function within pysam version 0.15.1 (https://github.com/pysam-developers/pysam), and normalized using the estimated size factors generated by DESeq2 with "normalized = T".

## PCA of genomic and transcriptomic responses

PCA using variants (SNPs) ascertained from the DNA read data was performed with all the pyflubumide-selected and paired control populations using SNPRelate version 1.18.0 [78]. The VCF file was filtered as described in [33] and transformed to the gds [78,79] format using the snpgdsVCF2GDS function with the option "method = copy.num.of.ref", at which point the snpgdsPCA function was used to construct the PCA with the "autosome.only = FALSE" option. A PCA based on the RNAseq read data was performed using the regularized log transformed counts.

## BSA genetic mapping

BSA genetic mapping was performed using methods adapted from previous studies [28,29,33], and specifically, using the defaults of the haplodiploid option of Kurlovs *et al.* [25]. Briefly, after quality control, JPR-R1 was specified as the haplodiploid male parent, and which alleles got passed on from its heterozygous sites (via the single haploid JPR-R1 male used in the BSA cross) was inferred using allele frequency information from the offspring; if the paired pyflubumide-selected and control populations had a frequency of 0.95 or higher, the site was considered non-segregating and not included in the final analysis. Sliding window analyses were performed using 75 kb windows with 5 kb offsets; at least 38 sites had to be present in a window for it to be included in the analysis. Significant responses to pyflubumide selection were detected using a permutation-based approach as described in Wybouw *et al.* [29]. The genome-wide allele frequency differences were permuted for the 11 pairs of selected/control populations $10^4$ times and were averaged across all pairs per permutation. For these analyses,

the three chromosomes were concatenated and made circular to maintain linkage information (the end of the third chromosome was connected with the beginning of the first chromosome). Using the $10^4$ permutations, the distribution of the absolute values of the maximal deviations in the averaged allele frequency differences was calculated and QTLs were assigned using an FDR of 0.05. BSA genetic mapping was also performed on the RNA read data using each set of paired BAM files. For each biallelic position in a coding exon where RNA read coverage was at least 20 and the major allele frequencies were less than 0.95 in the control populations, we calculated the allele frequencies for the two bases. The absolute allele frequency differences were calculated per paired set of pyflubumide-selected and susceptible populations and were plotted in averages of 500 kb non-overlapping sliding windows across the *T. urticae* genome [29]. Finally, the allele frequency differences were averaged across the replicated pairs. Protein domains were identified using Pfam- and CDD-searches [80,81].

## DNA quantitative PCR assay

RNA was reverse transcribed to cDNA using a RevertAid RT Reverse Transcription Kit (ThermoFisher Scientific). Primers 5'-CAAGGTGATCTCCAGCTTCA-3' and 5'-ACCGTTC GTGTATGCATGTT-3' were used to amplify *CPR*, yielding a 150 bp product. As a reference single-copy gene, we used the voltage-gated sodium channel (*VGSC*; *tetur34g00970*). Primers 5'-ATGATGCTTGGCAGTGAAAG-3' and 5'-ATAAGCAGCAGCAGCAAGAA-3' produced a 111 bp fragment of *VGSC*. PCR reactions were performed using Maxima SYBR Green/Rox qPCR Master Mix (Fermentas Life Sciences). All reactions were run in duplicate on an Agilent Technologies Stratagene Mx3005P thermocycler with the following conditions: an initial 10 min plateau at 95˚C, followed by 40 cycles that included denaturation at 95˚C for 15 sec, annealing at 55˚C for 30 sec, and extension for 30 sec at 72˚C. Melting curves were determined by an additional cycle of 95˚C for 1 min, 65˚C for 30 sec and 95˚C for 30 sec. One duplicate was excluded from the *VGSC* and *CPR* standard curves due to pipetting errors. Efficiencies were calculated from these standard curves and were incorporated in further calculations. First, the number of *VGSC* copies was determined by $2^*(\text{efficiency}^{(Ct-1)})$. Second, the Ct-values of *CPR* were corrected by log(efficiency)(*VGSC* copies/2)+1. The final estimate of *CPR* copy number was obtained with $\text{efficiency}^{(\text{corrected Ct–Ct})}$.

## Cytochrome P450 functional expression

We adopted a previously successful functional expression strategy for CYP392A16 [82]. Briefly, competent *E. coli* BL21STAR cells were co-transformed with the pCW_ompA-- CYP392A16 and pACYQ_TuCPR plasmids [82]. Transformed cells were grown at 37˚C in Terrific Broth under ampicillin and chloramphenicol selection until the optical density at 595 nm was higher than 0.95 cm$^{-1}$. The heme precursor δ-aminolevulinic acid was added to a final concentration of 1 mM. Induction was initiated by adding 1 mM of isopropyl-1-thio-β-D-galactopyranoside. After growing at 25˚C for 24 hours, cells were harvested and the spheroplasts were prepared. Membrane fractions were obtained from spheroplast solutions by ultracentrifugation at 180,000 g for 30 min at 4˚C. CYP392A16 membranes were diluted in TSE buffer (0.1 M Tris-acetate, pH 7.6, 0.5 M sucrose, 0.5 mM EDTA) and stored in aliquots at -80˚C. P450 content was measured by CO-difference spectra in reduced CYP392A16 membrane samples [83]. CPR activity was estimated by measurements of NADPH-dependent reduction of cytochrome c at 550 nm [84]. CYP activity was tested using the chemi-luminescent substrate Luciferin-ME EGE [82]. Total protein concentrations of CYP392A16 membranes were determined using the Bradford assay [85].

## Cytochrome P450 HPLC-UV activity assay

Pyflubumide and its carboxamide metabolite (NNI-0711-NH) (both 99.6% purity) were provided by Nihon Nohyaku (Japan). Pyflubumide and the toxic carboxamide were incubated at a concentration of 25 μM with 25 pmol of recombinant CYP392A16 in 100 μl Tris-HCl buffer (0.2 M, pH 7.4), containing 0.25 mM $MgCl_2$ and 2.5% acetonitrile (30˚C and 1250 rpm shaking). The reaction was performed in the presence and absence of an NADPH generating system: 1 mM glucose-6-phosphate (Sigma Aldrich), 0.1 mM NADP+ (Sigma Aldrich), and 1 unit/ml glucose-6-phosphate dehydrogenase (G6PDH; Sigma Aldrich). Reactions were stopped after 0 and 1 hr using 100 μl acetonitrile and further stirred for an additional 30 min. Finally, the quenched reactions were centrifuged at 10,000 rpm for 10 min and 100 μl of the supernatant was transferred to glass vials for HPLC analysis. Acaricides were separated on a UniverSil HS C18 (250mm 5um) reverse phase analytical column (Fortis). Reactions with pyflubumide and its NNI-0711-NH metabolite were separated using an isocratic mobile phase of 20% $H_2$0 and 80% acetonitrile with a flow rate of 1 ml/min for 20 min. Reactions were monitored by changes in absorbance at 260 nm and quantified by peak integration (Chromeleon, Dionex). For enzyme reaction kinetics, different concentrations of the NNI-0711-NH pyflubumide metabolite were used. Rates of substrate turnover from two independent reactions were plotted versus substrate concentration. $K_m$ and $V_{max}$ were determined using SigmaPlot 12.0 (Systat Sofware, London, UK).

## Identification of reaction products by HPLC-MS

Prior to HPLC-MS analysis, reaction mixtures were desalted by solid phase extraction (Bond Elute LRC-C18, 200 mg cartridges; Agilent, USA). Briefly, reaction mixtures were loaded to cartridges that were pre-conditioned with 3 mL 100% acetonitrile, followed by 3 mL 2.5% acetonitrile in water. Cartridges were subsequently washed with 1 mL water and samples were eluted with 1.5 mL acetonitrile. Eluents were transferred to HPLC autosampler vials and 500 μL of water was added to enhance chromatographic separation. Eluents were analyzed using an HPLC-MS/MS system. Sample injections (20 μl loop) were performed via a Surveyor Autosampler (Thermo Scientific, USA). Chromatographic separation was achieved using a Surveyor LCsystem (Thermo Scientific, USA), equipped with a Gemini C18 (3 mm, 100 mm x 2 mm) analytical column (Phenomenex, USA). An isocratic elution was applied with 80% acetonitrile-20% water and flow rate was set at 200 μL/min. Analyte detection was achieved using an electrospray ionization (ESI) triple quadrupole mass spectrometer (TSQ Quantum; Thermo Scientific, USA) operated in the positive ion mode. Mass spectrometry was operated both in full scan and product ion scan modes. The system was controlled by the Xcalibur software, which was also used for data acquisition and analysis. The optimum mass spectrometer parameters were set as follows: spray voltage at +4500 V, sheath gas pressure at 20 arbitrary units, auxiliary gas pressure at 10 arbitrary units, ion transfer capillary temperature at 300˚C and source collision induced dissociation at 26 eV. In the product ion mode the collision cell contained Argon at 1.5 mTorr. Sheath/auxiliary gas was high purity nitrogen and collision gas was high purity argon. For MS/MS analysis, collision energy was set at 25 eV.

## Supporting information

**S1 Fig. JPR-R1 allele frequencies in the segregating populations.** (A) Frequencies of JPR-R1 alleles in the susceptible (control) and pyflubumide-selected populations across the three QTL, as assessed in a sliding window analysis (see also Fig 4A). Chromosomes are ordered by decreasing length and are indicated by alternating shading. (B-D) Frequencies of JPR-R1 alleles in the susceptible (control) and pyflubumide-selected populations at QTL-1, QTL-2,

and QTL-3, respectively. Pyflubumide-selected populations that do not show near-fixation levels of JPR-R1 allele frequencies at the averaged BSA peaks are indicated by their identifier. For all panels, coverage is color-coded according to treatment (legend, top). Vertical lines indicate the locations of the averaged BSA peaks.
(PDF)

**S2 Fig. Average difference in JPR-R1 allele frequency in the susceptible (control) and pyflubumide-selected populations using RNA short-read data in a sliding window analysis.** Chromosomes are ordered by decreasing length and are indicated by alternating shading. Vertical lines indicate the locations of the three QTL, whereas vertical dashed lines indicate the two subsidiary peaks (Fig 4).
(PDF)

**S3 Fig. Transcript levels and genomic coverage of candidate genes.** *CYP392A16* (A) and *CPR* (B) expression in pyflubumide-selected populations relative to the mean RNA abundance in the susceptible control populations. (C) DNA coverage of *CPR* in pyflubumide-selected and control populations, relative to single-copy *VGSC*. DNA coverage was estimated by quantitative PCR analysis of the segregating populations, confirming estimates using short-read coverage (Fig 5). Panels are color-coded according to treatment.
(PDF)

**S4 Fig. Relative DNA coverage and gene models in ~75 kb genomic windows surrounding QTL-1 and QTL-2.** Vertical lines indicate the locations of averaged BSA peaks. Candidate genes *CYP392A16* (A) and *CYP392E6-8* (B) are highlighted in yellow. Otherwise, coding exons and introns are depicted as dark gray and lighter gray boxes, respectively. Symbols + and– denote forward and reverse gene orientations. Coverage is color-coded according to treatment and strain (legend, top).
(PDF)

**S5 Fig. Carbon monoxide difference spectra of membranes expressing *CYP392A16*.**
(PDF)

**S1 Table. The toxicity of pyflubumide in different susceptible and resistant strains and their crosses.**
(DOCX)

**S1 Data. Raw count data of the dose-response bioassays.**
(XLSX)

**S2 Data. Transcriptomic responses to pyflubumide selection.**
(XLSX)

## Acknowledgments

We thank Ludek Tikovsky and Harold Lemereis for their assistance in plant rearing and maintenance of greenhouse rooms, and Robert Greenhalgh for assistance with bioinformatic analyses. We are grateful to René Feyereisen for helpful comments. We sincerely thank Nihon Nohyaku for providing technical standards of pyflubumide and its NH metabolite.

## Author Contributions

**Conceptualization:** Seyedeh Masoumeh Fotoukkiaii, Nicky Wybouw, Thomas Van Leeuwen.

**Data curation:** Nicky Wybouw, Richard M. Clark, John Vontas, Thomas Van Leeuwen.

**Formal analysis:** Seyedeh Masoumeh Fotoukkiaii, Nicky Wybouw, Andre H. Kurlovs, Dimitra Tsakireli, Spiros A. Pergantis.

**Funding acquisition:** Nicky Wybouw, Richard M. Clark, John Vontas, Thomas Van Leeuwen.

**Investigation:** Seyedeh Masoumeh Fotoukkiaii, Nicky Wybouw, Andre H. Kurlovs, Dimitra Tsakireli, Spiros A. Pergantis.

**Project administration:** Thomas Van Leeuwen.

**Visualization:** Seyedeh Masoumeh Fotoukkiaii, Nicky Wybouw, Andre H. Kurlovs, Dimitra Tsakireli, Spiros A. Pergantis.

**Writing – original draft:** Seyedeh Masoumeh Fotoukkiaii, Nicky Wybouw.

**Writing – review & editing:** Seyedeh Masoumeh Fotoukkiaii, Nicky Wybouw, Andre H. Kurlovs, Dimitra Tsakireli, Spiros A. Pergantis, Richard M. Clark, John Vontas, Thomas Van Leeuwen.

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
