## [Decision Letter · Decision Letter 0]

15 Apr 2021

Dear Dr Wybouw,

Thank you very much for submitting your Research Article entitled 'High-resolution genetic mapping reveals cis-regulatory and copy number variation in loci associated with cytochrome P450-mediated detoxification in a generalist arthropod pest' to PLOS Genetics.

The manuscript was fully evaluated at the editorial level and by independent peer reviewers. The reviewers appreciated the attention to an important topic but identified some concerns that we ask you address in a revised manuscript

We therefore ask you to modify the manuscript according to the review recommendations. Your revisions should address the specific points made by each reviewer.

[LINK]

Yours sincerely,

Nadia D. Singh

Preprint Editor

PLOS Genetics

Kirsten Bomblies

Section Editor: Evolution

PLOS Genetics

The comments and suggestions from two expert reviewers are now available. Both reviewers agree that the study is exciting and contributes significantly to our understanding of the genetic basis of resistance in a major pest. The results are compelling and the writing is clear and concise. The reviewers have minor suggestions to improve clarity and expand the discussion someone. One additional thing the authors may wish to consider- if this is the first case of a CNV underlying resistance, the authors could amplify the novelty of this message somewhat.

Reviewer's Responses to Questions

**Comments to the Authors:**

Reviewer #1: The aim of the study is to identify the causes of resistance to a new acaricide, pyflubumide, in populations of Tetranychus urticae. The authors show that resistance is multifactorial, they identify three main QTLs that contain detoxification genes. Among these detoxification genes, a P450, CYP392A16, is capable of metabolizing the active deacylated form of the insecticide into a less toxic metabolite. This work represents a large body of results and is essential in understanding resistance mechanisms and managing a major pest. However, I believe that some improvements to the manuscript are possible, please see details below.

Major points

Results

- Paragraph “Candidate genes for pyflubumide resistance are linked to cytochrome P450-mediated detoxification”, from line 196 to 208, I am surprised that the authors didn’t do any comments on the gene “aryl hydrocarbon receptor interacting protein” which is down-regulated in QTL1. In mammals, this gene codes for a chaperone protein which binds to inactive Aryl Hydrocarbon Receptor (AhR) in the cell cytoplasm. AhR is known in arthropod to regulate the expression of detoxification genes. As a general comment for this manuscript, I would say that although I can understand that the authors have other works going on to characterize the cis-regulation involved in the up-regulation of P450s in their resistant populations, I found that discussion regarding this specific point is missing. Comparison between the promoter of CYP392A16 between susceptible and resistant population/mites is information available in author hands.

Discussion

- From line 324 to 329, in the same idea than the previous comment, the discussion regarding cis-regulation is very limited. Ok for “Future studies are needed to test the hypothesis of cis-regulation using a more suitable experimental set-up”, however the authors can at least say a few words on the already known mechanisms in arthropods (transcription factors involved).

Materials & methods

- Lines 397-398, “JPR-R1 and JPR-R2 were selected for high levels of pyflubumide resistance from the JP-R strain”, I understand that the authors don’t want to repeat what was already published, but I have to read the reference 41 to understand the difference between JPR-R1 and JPR-2. As most of the results of the manuscript are on these two strains, maybe the authors can add one more sentence to explain at least the selection way.

Figure 5: I find it frustrating not to have the names of the genes identified in the QTL regions (despite the fact that they are mentioned in the supplementary data S2). The authors could give names of P450 and CPR in smaller sizes and indicate the names of the other genes shown in the figure.

Supplementary Data S2

- In the genome-wide transcription, there are many detoxification genes given without a precise name. As far as I know all detoxification genes were annotated in the genome of Tetranychus urticae, so why is there still gene defined as: “Cytochrome P450”, “Carboxyl/cholinesterase”, etc….

Minor points:

- Line 293, please correct pyflubmide by pyflubumide.

- Line 631, correct meabolite by metabolite.

- Ref 9, add the five following names and et al. after Daborn PJ because he was not the sole author of this publication.

- Ref 19, the title appears two times, correct it.

- Ref 41, remove [cited 9 Mar 2020] and add volume and page number.

- Please recheck all the references (species names in italic, duplicated title, sometimes the name of the journal is in full sometimes in abbreviated form…).

Reviewer #2: This is a very nice study identifying the genetic determinants of pyflubumide resistance in spider mites, using genetic crosses, geomics, transcriptomics and functional expression. The methods are explained in detail and the results are convincing. I have only minor comments.

lines 177 and 484: In both the results and the methods, the method for obtaining a p-value for a given window is described only with a citation (and a mention, in the results, that it involves permutations). Please include a very brief description (one or two sentences) of the method.

lines 183-185: I found this sentence confusing, and it left me wondering how many "a number" was. I suggest rephrasing as "where the signal came close to significance", or replace "a number" with something a little more descriptive (eg: "most", or give respective numbers of replicates in which elevations were found).

lines 189-190: I don't understand the description of QTL genomic intervals at all. Please rephrase this more clearly. Please also state the size of each QTL genomic interval, as defined here.

lines 188-193: I do not understand how to interpret this result.. With the exception of large CNVs, I don't see why differentially-expressed genes should cluster within QTLs. Each QTL is presumably driven by one (or a few) mutations affecting the expression or sequence of just a handful of genes. Here, the authors argue that 123 genes DEGs are all clustered within these QTLs. At the end of the paragraph, the authors then argue that the 290 DEGs that do not fall within the 3 QTLs are trans-regulated from within the three QTLs, which would require yet more processes under selection, all falling within the same three regions. This all seems rather unlikely to me and, if true, it's not clear to me how this would occur. Would it be possible to include a statistical analysis of whether DEGs are significantly more likely to be found within a QTL than outside of one (the analysis would need to take into account the overall gene-richness of the QTL regions compared to the genome as a whole)? We are told that 30% of the DEGs are found within the QTLs, but what proportion of all genes are in the QTLs?

lines 202 and 211: the numbers of DEGs described for the intervals here are different to the numbers listed on lines 190-191, presumably because the "interval" used is different. I found this confusing again, perhaps because I could not undertand lines 189-190.

line 415: It seems to me that this is only the expected dose-response for monogenic resistance when resistance is completely recessive. I can't see a strong justification for using this as a fair test of the null hypothesis of monogenicity.

Figure 1: I don't understand what the "monogenic" lines are showing? As they the theoretical predictions from the equation on line 415?

**Have all data underlying the figures and results presented in the manuscript been provided?**

Reviewer #1: Yes

Reviewer #2: Yes

PLOS authors have the option to publish the peer review history of their article (what does this mean?). If published, this will include your full peer review and any attached files.

Reviewer #1: No

Reviewer #2: No

---

## [Editor Report · Decision Letter 1]

28 May 2021

Dear Dr Wybouw,

We are pleased to inform you that your manuscript entitled "High-resolution genetic mapping reveals cis-regulatory and copy number variation in loci associated with cytochrome P450-mediated detoxification in a generalist arthropod pest" has been editorially accepted for publication in PLOS Genetics. Congratulations!

Yours sincerely,

Nadia D. Singh

Preprint Editor

PLOS Genetics

Kirsten Bomblies

Section Editor: Evolution

PLOS Genetics

Comments from the reviewers (if applicable):

**Data Deposition**

http://datadryad.org/submit?journalID=pgenetics&manu=PGENETICS-D-21-00174R1

**Press Queries**

---

## [Editor Report · Acceptance letter]

16 Jun 2021

PGENETICS-D-21-00174R1 

High-resolution genetic mapping reveals cis-regulatory and copy number variation in loci associated with cytochrome P450-mediated detoxification in a generalist arthropod pest 

Dear Dr Wybouw, 

We are pleased to inform you that your manuscript entitled "High-resolution genetic mapping reveals cis-regulatory and copy number variation in loci associated with cytochrome P450-mediated detoxification in a generalist arthropod pest" has been formally accepted for publication in PLOS Genetics! Your manuscript is now with our production department and you will be notified of the publication date in due course.

With kind regards,

Katalin Szabo

PLOS Genetics

On behalf of:
